# 3D Printing of Drug Nanocrystals for Film Formulations

**DOI:** 10.3390/molecules26133941

**Published:** 2021-06-28

**Authors:** Giorgia Germini, Leena Peltonen

**Affiliations:** 1Department of Food and Drug Sciences, University of Parma, Parco Area delle Scienze, 27/A, 43124 Parma, Italy; giorgia.germini@gmail.com; 2Drug Research Program, Faculty of Pharmacy, University of Helsinki, P.O. Box 56, Viikinkaari 5E, 00014 Helsinki, Finland

**Keywords:** 3D printing, film formulation, nanocrystals, polymer, poor solubility, semi-solid extrusion, wet milling

## Abstract

The aim of the study was to prepare indomethacin nanocrystal-loaded, 3D-printed, fast-dissolving oral polymeric film formulations. Nanocrystals were produced by the wet pearl milling technique, and 3D printing was performed by the semi-solid extrusion method. Hydroxypropyl methyl cellulose (HPMC) was the film-forming polymer, and glycerol the plasticizer. In-depth physicochemical characterization was made, including solid-state determination, particle size and size deviation analysis, film appearance evaluation, determination of weight variation, thickness, folding endurance, drug content uniformity, and disintegration time, and drug release testing. In drug nanocrystal studies, three different stabilizers were tested. Poloxamer F68 produced the smallest and most homogeneous particles, with particle size values of 230 nm and PI values below 0.20, and was selected as a stabilizer for the drug-loaded film studies. In printing studies, the polymer concentration was first optimized with drug-free formulations. The best mechanical film properties were achieved for the films with HPMC concentrations of 2.85% (*w*/*w*) and 3.5% (*w*/*w*), and these two HPMC levels were selected for further drug-loaded film studies. Besides, in the drug-loaded film printing studies, three different drug levels were tested. With the optimum concentration, films were flexible and homogeneous, disintegrated in 1 to 2.5 min, and released the drug in 2–3 min. Drug nanocrystals remained in the nano size range in the polymer films, particle sizes being in all film formulations from 300 to 500 nm. When the 3D-printed polymer films were compared to traditional film-casted polymer films, the physicochemical behavior and pharmaceutical performance of the films were very similar. As a conclusion, 3D printing of drug nanocrystals in oral polymeric film formulations is a very promising option for the production of immediate-release improved- solubility formulations.

## 1. Introduction

Today, poor solubility creates great challenges in the drug industry, and nanosizing is an efficient and simple way to overcome the problem [1,2]. However, nanosizing is just the first step in the manipulation of raw drug material. After nanonization, production of the final formulation is still needed. Drug nanosuspensions can be formulated for oral drug delivery purposes, for example, for solid dosage forms, like tablets, capsules, granules, or the more novel oral polymeric films [3,4].

Oral polymeric films (also named oral thin films or oral strips) are a recent drug delivery form, which has high patient compliance, because they can be administered without water [5]. They can be produced, for example, by solvent casting [4], hot-melt extrusion [6], electrospinning [7], or 3D printing techniques [8], and for nanocrystal-based drug delivery systems, nanosuspensions can be used as such by only mixing the suspension with a polymeric excipient solution.

Fast-dissolving drug delivery systems have been studied since the late 1970s. They were invented in order to avoid swallowing problems with tablets, mainly for children or elderly populations. Oral polymeric films are formulated so to dissolve upon contact with moist surfaces, like mucus layers or tongue in the mouth. When the dissolution takes place in the mouth, drug degradation and first-pass metabolism can be avoided [9].

Oral polymeric films have been utilized for water–soluble drugs, but are also a good option for poorly water-soluble drugs [5]. The challenge with nanoparticle-based materials in polymeric films is, however, how to obtain homogeneous and mechanically performant films [10].

With the aid of 3D printing, solid, or, in some cases, semisolid objects, with various shapes, are produced through digitally controlled layer-by-layer material addition. In biomedical and pharmaceutical applications, the main interest in 3D printing has been in personalized medicine [11], but also bulk production is studied. The first drug formulation produced by 3D printing, approved by the FDA in 2015, was tablet formulation. 3D printing is indeed a group of different techniques, each having its own advantages and still presenting many open questions [12]. Their biomedical and pharmaceutical applications are numerous, such as tissue engineering [13], biopharmaceuticals [14], transdermals [15], tablet formulations with different geometries to fine-tune the drug release profiles [16], orodispersible formulations [17], just to mention a few.

Semi-solid extrusion 3D printers are suitable for printing hydrogel materials, for example, for polymeric film formulations, such as printed orodispersible films [13]. The benefits of this type of printing are the low printing temperature, allowing handling of even thermolabile drugs, and the use of disposable syringes, which can guarantee a high quality and purity of the end product [18]. In this type of printing, the viscosity of the printing ink is crucial for the end product properties, while the rheological properties reflect the printing performance [19]. For example, polymer concentration is related to the viscosity of the printing ink: when the viscosity of the feeding liquid/suspension is lowered, also the printing pressure is lowered [20,21]. When printing nanomaterials, incorporating nanosuspensions to the feed alters the rheological properties of the system, such as viscosity, which complicates the process further.

Successful ink-jet printing of drug nanosuspensions has been demonstrated [22], but, overall, 3D printing of drug nanosuspensions is rarely studied. However, 3D printing is a very convenient way to produce oral polymer film formulations, and nanocrystals can benefit from this type of final formulations. In our earlier study, we successfully produced drug nanocrystal-loaded oral polymeric films by the film casting method and showed that a thin-film formulation is a good option for drug nanocrystals [4]. In this study, we aimed to produce immediate-release formulations based on drug nanocrystals by the 3D printing method. The model drug, indomethacin, is a Biopharmaceutics Classification System (BCS) class 2 drug with poor solubility. The solubility of the drug material was improved by nanonizing it using the wet milling technique before the 3D printing process. The reasoning for selecting hydroxypropyl methyl cellulose (HPMC) as a film-forming polymer was dual: it is known to have good film-forming properties for thin-film formulations and it is also known to be a stabilizer for supersaturated drug solutions [23]. This is an important functional property for a system, when drug nanocrystals reach a supersaturated state after dissolution.

In this study, indomethacin nanocrystals were 3D-printed in order to make oral polymeric film formulations. The properties of the printed film were first screened using drug-free films. Based on these studies, the best HPMC concentrations for drug-loaded film studies were determined. In drug-loaded films, the effect of the amount of nanocrystalline drug on the properties of the final formulation was studied. Finally, for the best composition, 3D printing and film casting processes were compared. Thorough physicochemical and pharmaceutical characterization showed the 3D-printed nanocrystalline drug-loaded polymer films to be a promising option for immediate drug delivery purposes.

## 2. Results

### 2.1. Production of Drug Nanosuspensions

For drug nanosuspension production, three different stabilizers, HPMC and Poloxamers F127 and F68, were tested. The tested stabilizer concentrations (amount of stabilizers with respect to the drug amount, % (*w*/*w*)) were based on our earlier studies [24]: for HPMC, the stabilizer amount was selected to be 10% (*w*/*w*), for Poloxamer F127, 25% (*w*/*w*), and for Poloxamer F68, 60% (*w*/*w*) and 80% (*w*/*w*). Particle sizes and PDI values of the different tested indomethacin nanosuspensions are presented in Table 1.

Based on the particle size and PDI results, the smallest particles with the lowest polydispersity values were obtained with Poloxamer F68 as a stabilizer, and those two nanosuspension formulations were selected for further film studies.

### 2.2. Screening Studies for Film Formulations

Based on our earlier thin-film formulation studies, HPMC was selected as a film-forming polymer, and glycerol as a plasticizer [4]. Low-molecular-weight, meaning also low-viscosity, HPMC leads to a faster drug release; HPMC E5LV has in earlier studies shown to possess good film-forming properties [25].

The study was started by screening the optimum polymer concentration. The concentration level of the polymer was restricted by the fact that a highly viscose printing solution can lead to unwanted/uncontrolled bubble formation, which may cause dose variation problems. Besides, a high HPMC concentration can delay drug release via the gelling effect [26]. Based on earlier studies, the HPMC concentration of 2.2% (*w*/*v*) was selected as a starting point for the printing tests [4,27].

In film formulations, glycerol was used as a plasticizer. Plasticizers enhance the flexibility of the films and reduce their brittleness. Glycerol is a widely utilized plasticizer in pharmaceutical formulations and, as a small molecule, can easily penetrate between polymer chains, which controls the film properties. The presence of a plasticizer can also enhance the drug incorporation efficiency into a polymeric film [28]. The polymer/plasticizer ratio (5:1) was kept constant in all batches [4].

First, blank drug-free polymer films were tested with various polymer concentrations in order to optimize the film composition. One batch with only HPMC (2.2% (*w*/*v*)) was printed, and the other batches with higher HPMC amounts contained also glycerol. The HPMC concentration was gradually increased to 10% (*w*/*v*). The films were analyzed for the quality of their appearance, surface roughness, brittleness, and foldability without being ruptured. The appearance, flexibility, and cutting properties of the blank (drug-free) films are presented in Table 2.

When comparing the appearances of the printed films, all films were visually transparent and without any coloring. When the polymer concentration was increased, film thickness was also increased. Further, higher polymer concentrations led to rougher films, and the flexibility was reduced. HPMC 2.85% (*w*/*v*)–glycerol 0.57% (*w*/*v*) and HPMC 3.5% (*w*/*v*)–glycerol 0.7% (*w*/*v*) films displayed the best mechanical properties: they were soft enough and sufficiently resistant to bending and stretching. The thickness of these two films was lower but they were not brittle. Accordingly, these two HPMC–glycerol combinations were selected for further studies with drug nanosuspension-loaded films.

### 2.3. Indomethacin Nanosuspension-Loaded Films

Indomethacin nanosuspension-loaded film formulations were produced by the same 3D printing method as blank films. A high drug concentration (above 40–50% (*w*/*w*)) can lead to the formation of brittle films [29], and based on pre-testing, we selected the polymer/drug solution ratio of 2:1 throughout the study. HPMC and Poloxamer have been shown to be compatible with each other in film formulations; combined, they form homogeneous gels, which have good resistance towards erosion and good film-forming ability [30].

First, a fresh nanosuspension was directly mixed without any dilution with the polymer solution (keeping a constant solution ratio of 2:1). The resulting film was opaque and very brittle, indicating a too high solid particle concentration in the printing suspension. Hence, different dilutions of the nanosuspension were tested. Three solutions at three different dilutions of nanosuspension concentrations were prepared: 9% (*v*/*v*), 13% (*v*/*v*), and 26% (*v*/*v*). The most concentrated solution (26% *v*/*v*) was more brittle and more opaque than the diluted ones. In order to find the best film formulation, different dilutions of nanosuspensions, with two different stabilizer amounts (Poloxamer F68 60% and 80%) were printed with both combinations of HPMC/glycerol (2.85%/0.57% and 3.5%/0.7%) (Table 3).

#### Physicochemical and Mechanical Characterization of the Films

In our earlier studies, we showed that the produced indomethacin nanocrystal formulations, used in this study, are crystalline after the nanomilling [24]; hence, in this study, only the printed formulations were analyzed. In the thermograms, neither melting point peak of indomethacin nor glass transition in films was detectable, probably due to the relatively small amount of drug in the film formulation (Figure 1). The thermogram of the film formulation was very similar to that of pure HPMC, which reflected the high amount of HPMC in the final film formulation.

From the drug-loaded films, particle size after redispersion of the film pieces, visual appearance of the films, thickness, number of foldings before film breaking, weight variation of film pieces, drug content in the film pieces, and disintegration time were analyzed (Table 4).

The drug particle size of all the formulations was in the nanometer range, from 313 nm to 496 nm. The film formation process, both in 3D printing (films 1, 2 and 3) and in casting (film 4), slightly increased the particle size: the particle size in the printing/casting solution was appr. 225 nm. The appearance of all films was quite homogeneous. Some small air bubbles were visible in some films, and some aggregated drug spots were also seen in some films, which indicated uneven delivery of the drug in the film. Films were yellowish-white in color, and mostly opaque.

Films with the lowest amount of drug nanosuspension (1A, 1B, 1C, and 1D) had good folding endurance properties, but the weight variation and drug content uniformity were not good. The disintegration times were good for immediate-release formulations.

Films with the highest amount of nanosuspension (films 2B and 2C) were brittle, easily breakable, and did not endured folding. A too high drug amount caused poor endurance of the films. Cutting of these films was also difficult, the cut was not linear, and the films tended to break (rather than being cut) while cutting. It was clear that the drug content in these films was too high for good-quality films to be formed. The disintegration times of these films were the longest, and weight variations and variations in drug amount were high for these compositions.

We tested drug release in 30 min. All film formulations showed immediate drug release (Figure 2).

In all batches, after 10 min of dissolution, all the drug was released. Batches with a lower drug nanosuspension concentration (1A–D, 3A–C, and 4 A–C) reached 100% drug release already after 5 min. When 3D-printed (films 3A-C) and casted (films 4A–4C) films were compared, the casted films dissolved slightly faster, but the difference was not significant. Films with the highest nanosuspension concentrations (films 2B–2C) had slightly delayed drug release compared to those with lower drug amounts.

Based on a physicochemical and pharmaceutical analysis, a medium drug nanosuspension concentration (13% (*v*/*v*)) in the printing suspension seemed to be the best one. Using that concentration, both 3D printing as well as film casting were used to obtain drug nanosuspension-loaded films. As shown in Table 4, all these compositions formed durable films with short disintegration times (appr. 1–2 min). The film-forming process did not significantly change the final properties of the films, and the films were flexible and durable, with a good appearance. The weight variation and drug content uniformity were the best with these formulations, and drug release was immediate.

## 3. Discussion

In this study, a fast-disintegrating thin-film formulation for nanosuspension was developed. First, blank polymers and film plasticizers were studied. HPMC was the polymer of choice for this study, due to its good film-forming properties [4,25,31,32]. HPMC also stabilizes supersaturated states [33]. In drug nanosuspension formulations, where fast dissolution is followed by a supersaturated state, the presence of HPMC can be beneficial for inhibiting uncontrolled and unwanted precipitation; hence, it can also enhance the absorption. Glycerol was selected as a plasticizer, and it has been widely used in pharmaceutics in combination with HPMC [34]. It has also been shown that the presence of a plasticizer can produce more uniform and compact films [35].

For blank films, film-forming polymer (HPMC) concentrations from 2.2% (*w*/*v*) to 10% (*w*/*v*) were tested. Viscosity is an important factor affecting the film-forming ability. For HPMC E5 grade, a 2% aqueous HPMC solution has a viscosity of 5.02 mPas [36]. When the HPMC concentration is further increased, the viscosity is as well gradually increased: the viscosity of a 3% aqueous solution is 3 mPas, that of a 6% aqueous solution is 43 mPas, and that of an 8% aqueous solution is 91 mPas [37].

With higher HPMC concentrations, the films were more heterogeneous and harder and presented separated thicker and thinner areas and some bubbles. All films were easy to cut and quite flexible. The higher the HPMC concentration, the thicker the films. HPMC concentrations of 2.85% (*w*/*v*) and 3.5% (*w*/*v*) produced the most homogeneous and flexible films, and these concentration were selected for further drug-loaded film studies. These concentration were slightly higher than those found to be optimal in an earlier study (2.2%) [38].

Drug-loaded films were produced by mixing the aqueous HPMC/glycerol solution with the drug nanosuspension. In order to optimize the film composition, three different concentrations of the drug nanosuspension were tested.

All drug-loaded films were quite homogeneous and uniform in appearance, and only some minor aggregation spots were visible in some films. The higher drug loading led to easily breakable films with low folding endurance, and the film flexibility was lowered. Lower drug amounts in the film led to more flexible and durable films. In a study with a micronized drug, it has been noticed that acceptable polymer films were produced when the drug amount was below 30% (*w*/*w*) in the final composition [10]. Dispersed solid particles change the viscosity of the system and, hence, have an influence on the flow properties and, correspondingly, mechanical properties of films. The concentration in our study was close to the earlier reported limit, but in the earlier study, the particles were micronized, while we used nanonized particles [10]. This led to different behaviors in the suspension. For example, with nanonized particles, there is no sedimentation, but the aggregation tendency is higher.

The highest drug amount had a clear effect also on film thickness and disintegration time. The thickness with the highest nanosuspension concentration was from 100 nm to 110 nm, while with the lower nanosuspension concentrations, it was from 44 nm to 64 nm. The disintegration times with higher drug amounts were around 4 min, while with lower drug amounts, they were from 1 to 2.5 min. This difference also affected the drug release studies.

Particle size was measured after film redispersion. In all film formulations, the particle size after redispersion was from 313 nm to 496 nm. Though the particle size was slightly increased from the original particle size before film formation (225–227 nm), still, it was below 500 nm in all cases and clearly in the nanometer range, indicating that indomethacin nanocrystals were not irreversibly aggregated during the film formation process. For nanocrystal-based formulations, it is typical that the particle size in the final formulations is slightly increased as compared to that in freshly prepared nanosuspensions. For example, when nanocrystals were spray-dried, the particle size increased from 330 nm to values ranging from 360 nm to 560 nm [39]. The increase could be due to, for example, to excipient absorption on top of the particle surface or to moisture absorption. Similarly, the microenvironment around the particles is often changed due to altered measurement conditions (solution composition), which affects the measured hydrodynamic radius of the particles.

The highest nanosuspension concentration (26% in the printing ink) was clearly too high to obtain good-quality films. The drug concentration caused film breakability, lower resistance to folding, uneven drug loading to the film, and slow drug release rate. In 3D printing, the viscosity of the printing ink is an important parameter, and a high suspension concentration has an impact on that.

Based on the mechanical and pharmaceutical analysis, film formulations having the medium drug concentration (13% nanosuspension) were the best ones. With this composition, two different film-forming processes, namely, 3D printing and film casting, were compared. 3D-printed films showed less variation in the measured properties, though also casted films reached good results. With both techniques, the size of redispersed particle was from 310 to 420 nm. The main difference was in drug loading values, as printed films showed higher drug loading compared to casted ones, but the drug release was slightly faster for casted films (complete drug release in 2 to 3 min) as compared to printed films (3 to 5 min).

For the drug-loaded printed formulations, three different variables were studied, namely, drug amount, HPMC concentration, and stabilizer (Poloxamer) amount in the nanocrystal formulation. In this study, polymer and stabilizer concentrations did not have a real impact on the final product’s properties. The HPMC concentration was already optimized in the drug-free formulation studies, and the differences between the two tested levels were very small. In the same way, two different tested Poloxamer concentrations produced very similar particle size values.

As a conclusion, in this study, fast-dissolving oral polymeric film preparations loaded with drug nanocrystals were successfully produced by the 3D printing method. Drug loading level as well as polymer concentration impacted the final product’s properties. When the best film compositions made by 3D printing were compared to the same compositions prepared by film casting, both techniques resulted to produce high-quality films. The main differences between these two film formulations were in their drug loading and drug release properties, though the differences were very small.

## 4. Materials and Methods

Indomethacin was used as a poorly soluble model drug (Tokyo Chemical Industry Co., Tokyo, Japan). Poloxamer 188 (F68, Sigma-Aldrich Chemie GmbH, Steinheim, Germany) and Poloxamer 407 (F127, Sigma-Aldrich Chemie GmbH, Steinheim, Germany) were used as stabilizers, hydroxypropyl methylcellulose (HPMC) (E5 Premium LV, The Dow Chemical Company, Midland, Michigan, USA) as a film-forming polymer and also as a stabilizer in the milling studies, and glycerol (glycerol 85%, Oriola, Espoo, Finland) as a plasticizer. Water used was ultrapurified Milli-Q-water (Millipore SAS, Molsheim, France).

Indomethacin nanosuspensions were prepared by the wet milling technique in a planetary ball mill (Pulverisette 7 Premium, Fritsch Co., Idar-Oberstein, Germany) [24]. Parameters for the milling were as follows: pearl size Ø 1 mm, drug amount 1 g, milling speed 1100 rpm, total milling time 30 min (10 times 3 min milling, and after each 3-min milling, a milling vessel was placed into an ice bath in order to cool it down for 10 min).

Mean particle size and polydispersity index (PDI) values of the nanoparticles were analyzed by Photon Correlation Spectroscopy (PCS) (Malvern Zetasizer 3000HS, Malvern Instruments, Malvern, UK). Particle size information was collected from freshly prepared nanosuspensions as well as from 3D-printed and casted films after redispersion of the films. For redispersion, 0.9 × 0.9 cm^2^ sized film pieces were cut and dispersed in 5 mL of water. Before particle size analysis, 2 min sonication was performed.

The film-forming polymer solution for 3D printing was prepared by dissolving HPMC and glycerol into water. For the printing gel solution, drug nanosuspensions were diluted with water to three different concentrations, namely, 9% (*v*/*v*), 13% (*v*/*v*), and 26% (*v*/*v*). Polymer solution: the nanosuspension ratio was kept constant (2:1, i.e., two parts of polymer solution (12 mL) and one part of diluted drug nanosuspension (6 mL)).

Printing was performed with a Multitool 3D Printer, (ZMorph VX, Wroclaw, Poland) with thick paste extruder—option with a syringe. Before filling the syringe, a small piece of Parafilm was placed on the top of the lid in order to prevent the liquid from running through. Then, the syringe was filled with the printing solution (at least 20 mL), closed, and set into the printer. A spherical film print with the diameter of 0.7 mm was produced in a Petri dish. The printing parameters were as follows: layer height 2 mm, path width 1 mm, printing speed 10 m/s, paste thickness 5%.

Solvent film casting was performed with the same solutions as printing, but the solution (9 mL) was just poured in a Petri dish.

After printing and casting, the films were dried for 14–18 h in an oven at 40 ± 1 °C.

All the films were characterized in appearance, folding endurance, thickness, weight variation, drug content uniformity, disintegration, drug release, and physical state of the drug. For all film characterization analyses, the films were cut to 0.9 × 0.9 cm^2^ pieces.

Appearance was detected by visual and microscopical observation (Leica microsystems CMS GmbH, Bensheim, Germany).

Folding endurance was tested by calculating the number of repetitive foldings stressing the same specified area of the film until cracks were noticed.

Film thickness was determined by a digital micrometer (Sony Magnescale Inc., DZ521, Japan). Measurements were repeated for each sample in 5 randomly selected different positions, and the average thickness was calculated.

For the weight variation test, 3 pieces were randomly selected for each film, and the average weight was calculated.

For the content uniformity test, for each film, 3 randomly selected film pieces were tested. Each sample was placed into a volumetric flask and dissolved in 50 mL of ethanol. Drug amount was determined by a UV–Vis spectrophotometer (UV-1600PC, VWR International, Leuven, Belgium) at 318 nm wavelength.

The disintegration test was performed in Petri dishes. For each film, 6 different pieces were tested by putting each of them separately into a Petri dish containing 20 mL of water. The dishes were stirred at 10 s intervals. Film disintegration was detected visually.

Drug release studies were performed with the pharmacopeial paddle method in potassium phosphate buffer (pH 6.8, 500 mL). For each film, 3 pieces were tested. A 3 mL sample was withdrawn at preselected time intervals (0.5, 1, 1.5, 2, 3, 5, 15, and 30 min) and was replaced with the same amount of fresh buffer. The drug concentration was determined by a UV–Vis spectrophotometer at 318 nm wavelength.

The physical state of the drug was studied by differential scanning calorimetry (DSC 823e, Mettler Toledo, Columbia, USA). Besides the formed films, pure indomethacin, poloxamer, and HPMC were analyzed. For the analysis, samples were put into aluminum pans, which were closed with a perforated cap. The scanning rate was 5 °C/min, from 25 °C to 200 °C. Measurements were performed under a nitrogen gas flow (50 mL/min).

## Figures and Tables

**Figure 1 molecules-26-03941-f001:**
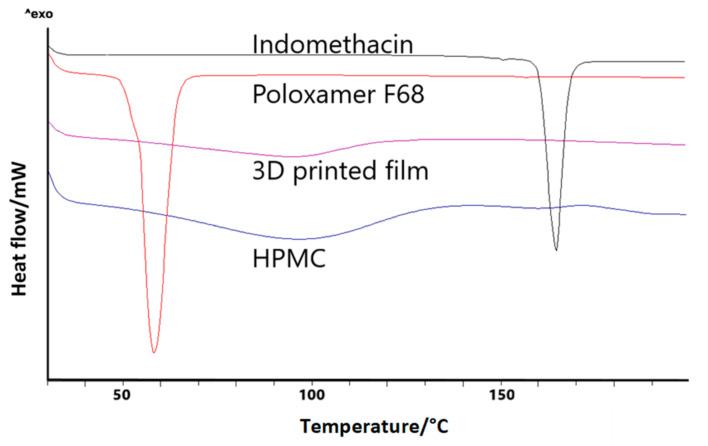
DSC thermograms of indomethacin, Poloxamer F68, HPMC, and 3D-printed film (exo up).

**Figure 2 molecules-26-03941-f002:**
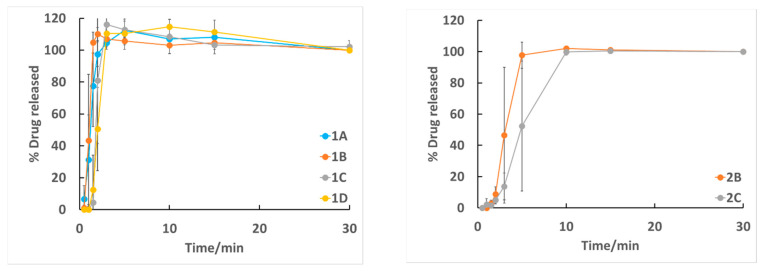
Drug release profiles of different film formulations (n = 3).

**Table 1 molecules-26-03941-t001:** Size and PDI information of the produced indomethacin nanosuspension batches. (The amount of stabilizer is given in *w*/*w* percentages with respect to the amount of drug).

Stabilizer	F127 25%	HPMC 10%	F68 60%	F68 80%
Particle size/nm	295.9 ± 1.2	734.7 ± 1.6	239.1 ± 1.8	228.6 ± 3.7
PDI	0.248 ± 0.060	0.232 ± 0.033	0.200 ± 0.011	0.148 ± 0.011

**Table 2 molecules-26-03941-t002:** Appearance, flexibility, and cutting properties of blank films. HPMC and glycerol concentrations are expressed in % (*w*/*v*), and their ratio was kept constant (5:1) throughout the tests.

Composition	Appearance	Flexibility	Cutting
HPMC 2.2%	No bubbles, little folds, low thickness	Flexible film, does not break by folding, brittle, quite elastic, slightly deformable before breaking	Easy to cut, does not break
HMPC 2.85% Glycerol 0.57%	Homogeneous film, little folds, low thickness	Flexible, resistant to bending, softer than HPMC 2.2% film, not brittle	Easy to cut, linear cut, no ripples
HMPC 3.5% Glycerol 0.7%	Homogeneous film, some little folds, no bubbles	Flexible, resistant to bending, not brittle, no cracks	Easy to cut, linear cut, no ripples
HMPC 4%, Glycerol 0.8%	Film has some little bubbles, medium thickness, little folds	Flexible, does not break by bending	Easy to cut, does not break, linear cut, no ripples
HMPC 5% Glycerol 1%	Film is wavy (little folds), only few little bubbles, thicker than the films with less HPMC	Flexible, does not break by folding	Easy to cut, harder than the films with a lower amount of HPMC, does not break
HMPC 6%, Glycerol 1.2%	Film has some little bubbles, thicker than the films with a lower amount of HPMC, heterogeneous, has thicker and thinner areas	Tends to break by folding, quite resistant to bending	Easy to cut, does not break, harder than the films with a lower amount of HPMC
HMPC 10% Glycerol 2%	No folds, some little bubbles, thicker and harder as compared to all the other films	More plastic, not elastic, breaks easily when bended	Easy to cut, does not break, hardest film

**Table 3 molecules-26-03941-t003:** Composition of the different tested film samples for printing (films 1A–3C). As a reference sample, film casting was performed with a 13% nanosuspension concentration (films 4A–4C).

	Nanosuspension F68 60%		Nanosuspension F68 80%	
Nanosuspension Concentration	HPMC 2.85% Glycerol 0.57%	HPMC 3.5% Glycerol 0.7%	HPMC 2.85% Glycerol 0.57%	HPMC 3.5% Glycerol 0.7%
9%	1A	1B	1C	1D
26%		2B	2C	
13% (printed)	3A	3B	3C	
13% (casted)	4A	4B	4C	

**Table 4 molecules-26-03941-t004:** Characteristics of the drug-loaded films: particle size values measured for the printing suspensions before printing and after film redispersion, appearance of the films, film thicknesses, folding endurance (number of foldings before the film is broken), weight variations, drug amounts, and disintegration time results (n = 3–6).

Film	Particle Size/nm		Appearance	Thickness/µm	Folding Endurance	Weight Variation/mg	Drug Amount/µg	Disintegration Time/s
	Nanosuspension	Dispersed Film						
1A	227.9 ± 1.8	346.5 ± 6.9	Uniform film, few little bubbles, no folds, flexible, not brittle	51 ± 8	4 ± 1	5.45 ± 1.23	616 ± 151	101 ± 1
1B	227.9 ± 1.8	428.6 ± 1.8	Uniform film, few little bubbles, no folds,	44 ± 5	4 ± 1	5.22 ± 0.34	544 ± 44	78 ± 1
1C	225.4 ± 1.4	400.5 ± 8.4	Uniform film, no bubbles no folds, flexible, not brittle	47 ± 11	3 ± 1	5.58 ± 1.27	638 ± 166	84 ± 2
1D	225.4 ± 1.4	405.0 ± 12.3	Mostly uniform film with little aggregation spots, little folds, no bubbles	55 ± 16	4 ± 1	7.16 ± 2.25	714 ± 235	101 ± 2
2B	227.9 ± 1.8	496.3 ± 10.8	Homogeneous film, no folds, small aggregation spots, only single small bubbles, thick, not brittle	100 ± 15	2 ± 1	10.08 ± 1.34	2266 ± 549	234 ± 1
2C	225.4 ± 1.4	418.2 ± 15.4	Homogeneous film, no folds, no visible aggregation, no bubbles, thick, not brittle	110 ± 25	1 ± 1	10.42 ± 1.70	2804 ± 824	232 ± 1
3A	227.9 ± 1.8	313.2 ± 9.9	Considerably homogeneous film, some aggregation spots, no bubbles	58 ± 6	4 ± 1	6.10 ± 0.07	1003 ± 15	108 ± 1
3B	227.9 ± 1.8	378.6 ± 3.5	Almost homogeneous, very few small aggregation spots, no bubbles	64 ± 5	4 ± 1	6.87 ± 0.45	945 ± 74	153 ± 1
3C	225.4 ± 1.4	363.7 ± 11.5	Almost homogeneous film, some aggregation spots	64 ± 8	3 ± 1	5.92 ± 0.70	1145 ± 151	110 ± 1
4A	227.9 ± 1.8	339.4 ± 5.8	Homogeneous film, little bubbles, no folds, few aggregation spots	46 ± 5	3 ± 1	5.26 ± 0.12	727 ± 28	76 ± 1
4B	227.9 ± 1.8	418.3 ± 8.9	Homogeneous film, little bubbles, no folds, few aggregation spots	59 ± 8	3 ± 1	5.92 ± 0.52	690 ± 94	111 ± 1
4C	225.4 ± 1.4	337.4 ± 6.7	Most homogeneous film, no bubbles, no folds, few aggregation spots	55 ± 12	3 ± 1	6.08 ± 1.15	1012 ± 227	68 ± 1

## Data Availability

The data presented in this study are available on request from the corresponding author.

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
