# Peer review of "3D Printing of Drug Nanocrystals for Film Formulations"

_molecules, 2021, doi:10.3390/molecules26133941_

Round 1
Reviewer 1 Report
The submitted work is a nice application of 3D-printing technology to the preparation of polymeric films loaded with an API.
The paper is well written and the presented experiments are well-designed and clearly discussed.
For these reasons I suggest to accept the manuscript in its present form.
Author Response
Thank you very much for the positive response. The manuscript has been read through carefully and English language check has been performed.
Reviewer 2 Report
The authors prepared the 3D printed polymeric films incorporated with indomethacin nanocrystal loaded. The authors claim that the pharmaceutical performance of the 3D printed drugs is same as the traditionally casted films. The 3D printed drug nanocrystals showed promising results in regards of drug loading and drug release. I would suggest revising and addressing the following issues in the manuscript.
- There is need to review the manuscript for English language corrections and typographical errors.
- In table 2 and elsewhere, only the appearance of the printed films is mentioned. The microscopic studies are performed but it is recommended that the films should be shown in image/photo form, wherever possible for better visible understanding to the reader.
- It is mentioned in the manuscript that the HPMC concentration was started from 2.2% (w/v) based on previous studies, but as viscosity is an important factor in 3D printing it was expected to perform the rheological studies for better understanding of the viscosity nature of printing samples.
- Redraw the figure 1. The resolution is does not correct. The figure labels are not neat and clean.
- Why the advanced microscopic characterization studies such as SEM and TEM are not performed to investigate the distribution of indomethacin nanocrystal in films?
Author Response
The authors prepared the 3D printed polymeric films incorporated with indomethacin nanocrystal loaded. The authors claim that the pharmaceutical performance of the 3D printed drugs is same as the traditionally casted films. The 3D printed drug nanocrystals showed promising results in regards of drug loading and drug release. I would suggest revising and addressing the following issues in the manuscript.
Comment: There is need to review the manuscript for English language corrections and typographical errors.
Answer: We apologize the language and typographical errors. The manuscript has carefully been reviewed in order to improve the language and to remove the typographical errors.
Comment: In table 2 and elsewhere, only the appearance of the printed films is mentioned. The microscopic studies are performed but it is recommended that the films should be shown in image/photo form, wherever possible for better visible understanding to the reader.
Answer: The film thicknesses were such that the microscopical imaging could be used only for supporting the visual inspection. We tried to take pictures from the microscopical figures, but they were not successful due to the thickness of the film, and that is why no images are attached to the manuscript. The optical microscope was utilized only in side of the visual inspection. We agree that the images could help to visualized more clearly the information in the table 2, but unfortunately the images were not good enough in quality to be added to the publication.
Comment: It is mentioned in the manuscript that the HPMC concentration was started from 2.2% (w/v) based on previous studies, but as viscosity is an important factor in 3D printing it was expected to perform the rheological studies for better understanding of the viscosity nature of printing samples.
Answer: Thank you for the very relevant comment. The viscosity information of different concentrations of aqueous HPMC solutions has now been added to the manuscript (rows 264-269, highlighted with yellow color).
Comment: Redraw the figure 1. The resolution is does not correct. The figure labels are not neat and clean.
Answer: The figure 1 has been withdrawn and the resolution and quality of the figure is improved. We hope that the quality of the figure is now high enough.
Comment: Why the advanced microscopic characterization studies such as SEM and TEM are not performed to investigate the distribution of indomethacin nanocrystal in films?
Answer: Thank you for the very good suggestion. The main aim in characterization of the films was to control the performance of the nanocrystals and to check that the nanocrystals are not aggregated irreversibly in the film. The redispersion studies were performed in order to show that the nanoparticles are still forming nanosuspensions after the films were dissolved. Also drug release studies with immediate drug release show that no permanent aggregation to microparticles are taken place in the films. SEM studies might have given more information about the surface properties of the films. TEM analysis might have revealed information related to the impeding of the nanoparticles into the cellulose matrix. But the aggregation is only facing the situation in the dry film state, not facing the fact, how the aggregates are behaving while in contact with water. In this study, the main aim was on the functionality of the film formulations, eg. the dissolution and redispersion studies related to particle size analysis was seen more important in this study. But the TEM/SEM studies are absolutely worth to study in the future.
Round 2
Reviewer 2 Report
After carefully reading the review comments answers I would recommend that the manuscript can be accepted in present form.